# Dynamics of Hydrology and Anaerobic Hydrocarbon Degrader Communities in A Tar-Oil Contaminated Aquifer

**DOI:** 10.3390/microorganisms7020046

**Published:** 2019-02-09

**Authors:** Giovanni Pilloni, Anne Bayer, Bettina Ruth-Anneser, Lucas Fillinger, Marion Engel, Christian Griebler, Tillmann Lueders

**Affiliations:** 1Institute of Groundwater Ecology, Helmholtz Zentrum München—German Research Center for Environmental Health, 85764 Neuherberg, Germany; giovanni.pilloni@exxonmobil.com (G.P.); anne.bayer@lfu.bayern.de (A.B.); bettina.ruth-anneser@fh-rosenheim.de (B.R.-A.); lucas.fillinger@helmholtz-muenchen.de (L.F.); christian.griebler@univie.ac.at (C.G.); 2Research Unit Comparative Microbiome Analysis and Research Unit Scientific Computing, Helmholtz Zentrum München—German Research Center for Environmental Health, 85764 Neuherberg, Germany; marion.engel@helmholtz-muenchen.de

**Keywords:** BTEX, anaerobic toluene degradation, benzylsuccinate synthase, microbial community dynamics, next-generation sequencing

## Abstract

Aquifers are typically perceived as rather stable habitats, characterized by low biogeochemical and microbial community dynamics. Upon contamination, aquifers shift to a perturbed ecological status, in which specialized populations of contaminant degraders establish and mediate aquifer restoration. However, the ecological controls of such degrader populations, and possible feedbacks between hydraulic and microbial habitat components, remain poorly understood. Here, we provide evidence of such couplings, via 4 years of annual sampling of groundwater and sediments across a high-resolution depth-transect of a hydrocarbon plume. Specialized anaerobic degrader populations are known to be established at the reactive fringes of the plume. Here, we show that fluctuations of the groundwater table were paralleled by pronounced dynamics of biogeochemical processes, pollutant degradation, and plume microbiota. Importantly, a switching in maximal relative abundance between dominant degrader populations within the Desulfobulbaceae and *Desulfosporosinus* spp. was observed after hydraulic dynamics. Thus, functional redundancy amongst anaerobic hydrocarbon degraders could have been relevant in sustaining biodegradation processes after hydraulic fluctuations. These findings contribute to an improved ecological perspective of contaminant plumes as a dynamic microbial habitat, with implications for both monitoring and remediation strategies in situ.

## 1. Introduction

Over 97% of global non-glacial freshwater resources are retained in groundwater [1], which constitutes the main source of drinking water in many areas of the world. Despite this key relevance for our society, aquifers are still poorly understood as ecosystems, especially regarding the ecology of their primary populace, the microbes [2]. In contrast to many other aquatic ecosystems, aquifers have been classically perceived as mostly steady-state habitats, not subject to pronounced diurnal or seasonal dynamics, and characterized by a rather stable hydrology, redox, temperature, as well as availability of energy and nutrients [3,4]. Even after contamination, for example, by petroleum hydrocarbons, steady-state contaminant plumes and respective anoxic redox compartments are understood to establish and to be attenuated over time scales of decades to centuries [5]. While contamination clearly represents a fundamental perturbation of aquifer microbes and ecosystem status [6,7,8,9,10], a better understanding of the ecology of contaminant plumes is needed to improve site-specific prediction and restoration strategies [11].

At a tar-oil contaminated aquifer in Düsseldorf-Flingern in Germany, we have previously shown that a locally enriched population of specialized anaerobic degraders had established at the sulfidogenic lower fringe of a toluene plume, indicative of a “hot-spot” of biodegradation in situ [12,13,14,15]. This was consistent with the hypothesis that degraders are primarily controlled by the limited dispersive mixing of electron donors and acceptors in porous media, thus largely restricting their activities to the reactive fringes of contaminant plumes [16]. To further elaborate the importance of such plume fringes for overall biodegradation, it is relevant to understand them as microbial habitats, and to elucidate possible ecological factors limiting the establishment and activity of degrader populations. Especially, possible feedbacks between abiotic habitat dynamics, such as fluctuating groundwater tables and degrader populations, have been rarely addressed to date.

It has been previously demonstrated that fluctuating groundwater levels can drive dynamics in contaminant distribution and related biogeochemical processes [17,18,19]. Moreover, Haack et al. observed general rearrangements of aquifer bacterial communities and a loss of diversity in distinct zones of a hydrocarbon plume after hydraulic dynamics [20]. However, the potentially complex interplay between hydraulic and microbial parameters in groundwater is still considered as one of the key unknowns in aquifer restoration today [21]. Most of the aforementioned studies have sampled microbes from groundwater and not from sediments [17,18,19,20], where spatially explicit depth-resolved interpretation is more straightforward [11,22]. In the present study, we hypothesize that anaerobic degrader communities established at the reactive fringe of the previously characterized toluene plume could respond to hydraulic habitat fluctuations by community rearrangements. We follow up on this via four years of repetitive depth-resolved groundwater and sediment sampling at the tar-oil contaminated Flingern aquifer, as well as comprehensive hydrogeochemical, isotopic, and microbial community analyses.

## 2. Materials and Methods

### 2.1. Site Sampling and Hydrochemical Analyses

Sediment and groundwater were sampled at a well-studied tar-oil contaminated aquifer in Düsseldorf-Flingern (Germany) [12,13,14,15]. The aquifer is dominated by medium and coarse sands, with thin interspersed gravel layers at depths below 10 m. The hydraulic conductivity of the aquifer is ~10^−3^ m·s^−1^, with a mean groundwater flow velocity at ~1 m·d^−1^ [23]. The intrinsic hydrocarbon plume is dominated by toluene (~60–80% of total petroleum hydrocarbons) and sulfidogenic degradation [12,13,14,15]. Water samples from a high-resolution multi-level well (HR-MLW) were collected between ~5.5 and 8.5 m depth in February 2006 [13], February 2007, September 2008, and June 2009. For relevant plume compartments, depth-resolved monitoring was conducted with a depth resolution as little as 5–10 cm. In part, data from a previous sampling in September 2005 [12,24] are also included in the present study. Hydrogeochemical parameters were analyzed from water samples as previously described [14,24]. In addition, toluene δ^13^C/^12^C isotope ratios were analyzed from duplicate NaOH-preserved samples with gas chromatography-combustion-isotope ratio mass spectrometry (GC-C-IRMS, Thermo Fisher, Bremen, Germany) via purge and trap analysis [25] and gas chromatographic separation adapted to benzene, toluene, ethylbenzene, and xylenes (BTEX). A DB624-column (Supelco, Bellefonte, PA, USA) and an optimized temperature program (3.07 min at 60 °C, 8.1 °C/min to 129 °C, 1 °C/min to 134 °C, 20 °C/min to 180 °C, 50 °C/min to 230 °C, hold 5 min) were used. The standard deviations for duplicate toluene δ^13^C/^12^C isotope measurements were much lower than the instrumental uncertainty and were therefore set to 0.5‰ as recommended [26].

At matching time points in 2006, 2007, and 2009, water samples were also taken by the site owner (SWD, Stadtwerke Düsseldorf, Germany) from the surrounding network of conventional monitoring wells to assess the horizontal extent of the plume. For these samples, concentrations of BTEX compounds were quantified following standardized methods (DIN 38407-F9:1991-05) by a certified analytical laboratory of the SWD. Plume extent was interpolated by Kriging by the SWD for samples taken in 2006 and 2007, as well as for 2009 by BFM-Umwelt (Munich, Germany) using Surfer 9.0 (Golden Software Inc., Golden, CO, USA).

For sediment sampling, intact sediment liners were taken as direct-push cores within distances of 0.5 to 1 m next to the HR-MLW in February 2006, September 2008, and June 2009. Sediment liners were retrieved with a direct-push drilling rig (Geoprobe, Salina, KS, USA) and shock-frozen on dry ice directly after retrieval. Per drilling, eight sediment depths representing all major plume compartments over a depth range between 5.5 and 8.5 m below ground were sub-sampled as micro-cores from the sediment liners with cut sterile syringes, then handled and analyzed as described [12,13].

### 2.2. Molecular Analyses

DNA templates were extracted from triplicate frozen sediment sub-samples of each depth within 6–12 months after sampling. DNA was extracted from sediment aliquots suspended in 650 mL PTN buffer (120 mM Na_2_HPO_4_, 125 mM Tris, 0.25 mM NaCl (pH 8)) and incubated at 37 °C for 15 min with 40 mL lysozyme (50 mg·mL^−1^) and 10 mL proteinase K (10 mg·mL^−1^). After adding 150 mL 20% (*wt*/*v*) sodium dodecyl sulfate, the incubation was prolonged for 15 min at 65 °C in a shaker at 500 rpm. The slurries were then bead beaten (45 s at 6.5 ms^−1^) in a FastPrep-24 (MP Biomedicals, Solon, OH, USA) with 0.2 mL of zirconia-silica beads (1:1 mix of 0.1-mm and 0.7-mm in diameter; Roth, Karlsruhe, Germany) and 100 mL of phenol-chloroform-isoamyl alcohol (25:24:1) in 2-mL screw-cap vials. Subsequently, the nucleic acids were sequentially purified by extraction with 1 volume of phenol-chloroform-isoamyl alcohol (25:24:1) and 1 volume of chloroform-isoamyl alcohol (24:1). Purified nucleic acids were then precipitated with 2 volumes of 30% polyethylene glycol by incubation at 4 °C for at least 2 h and subsequently centrifuged at 20,000× *g* and 20 °C for 30 min. All chemicals used in the DNA extraction were from Sigma-Aldrich (St. Louis, MO, USA) unless otherwise specified. For each biological sample, two parallel sediment extractions were pooled in 60 mL of elution buffer and stored at −20 °C until further analysis. Extracted nucleic acids were first subjected to terminal restriction fragment length polymorphism (T-RFLP) fingerprinting of bacterial 16S rRNA genes, as previously described [27]. Amplicons for T-RFLP were generated using the 5′-FAM-labelled primer Ba27f (5′-aga gtt tga tcm tgg ctc ag-3′) and the unlabeled primer 907r (5′-ccg tca att cct ttg agt tt-3′) in a Mastercycler ep gradient (Eppendorf, Hamburg, Germany) with the following cycling conditions: initial denaturation (94 °C, 5 min), followed by 24 or 28 cycles of denaturation (94 °C, 30 s), annealing (52 °C, 30 s), and elongation (70 °C, 60 s). Each 50 mL PCR reaction contained 1× PCR buffer, 1.5 mM MgCl_2_, 0.1 mM dNTPs, 1.25 U recombinant Taq polymerase (all from Fermentas, St. Leon-Rot, Germany), 0.2 mg mL^−1^ bovine serum albumin (BSA) (Roche, Penzberg, Germany), 0.5 mM of each primer (Biomers, Ulm, Germany), and 1 µL of template DNA. Amplicons were restricted using *Msp*I and separated by capillary electrophoresis.

Quantitative PCR (qPCR) of anaerobic toluene degradation genes (benzylsuccinate synthase α-subunit, *bssA*) and total bacterial 16S rRNA genes was also done as previously described [12]. The utilized *bssA* detection assay was specific for the Desulfobulbaceae-related “F1-cluster” *bssA* previously shown to dominate at the site [12]. *BssA* qPCR was performed with a TaqMan universal master mix kit (Applied Biosystems) with the primers bssApd2f (5′-cct atg cga cga gta agg tt-3′) and bssApd2r (5′-tga tag caa cca tgg aat tg-3′) used in combination with the probe bssApd2h (5′-tcc tgc aaa tgc ctt ttg tct caa-3′). A thermal cycle of initial denaturation at 95 °C for 10 min was followed by 50 cycles of denaturation at 95 °C for 15 s, annealing at 55 °C for 20 s, and final elongation at 72 °C for 30 s. Total bacterial 16S rRNA genes were quantified using a previously described Sybr green PCR approach using the primers Ba519f (5′-cag cmg ccg cgg taa nwc-3′) and Ba907r. We used standard Taq polymerase (Fermentas) assays in the presence of 0.1× Sybr green (FMC Bio Products, Philadelphia, PA, USA) and 2 μl DNA template. Initial denaturation (94 °C, 3 min) was followed by 50 cycles of denaturation (94 °C, 15 s), annealing (52 °C, 15 s), and elongation (70 °C, 30 s). Both qPCRs were performed on a MX3000P qPCR cycler (Stratagene, La Jolla, CA, USA). For each sediment depth, the three biological DNA extracts were quantified in three different dilutions (undiluted, 1:5, and 1:10) to account for the possibility of PCR inhibition.

After initial T-RFLP and qPCR screening, 16S rRNA gene amplicon pools from five depths between 6.4 and 8.4 m below ground were chosen for each year and subjected to amplicon pyrosequencing on a FLX Genome Sequencer (Roche—454 Life Sciences, Branford, CT, USA). Sequencing was done from deep-frozen DNA extracts (−80 °C) using Titanium chemistry (Roche) as previously described [28,29]. Amplicons for multiplexing were prepared with the primers Ba27f and Ba519r (5′-tat tac cgc ggc kgc tg-3′) extended as amplicon fusion primers with respective primer A or B adapters, key sequences, and multiplex identifiers (MIDs). Sequence data denoising, quality trimming, and chimera checking was performed using mothur v.1.34.3 [30] as previously described [29]. Subsequently, sequences were classified using the SILVAngs data analysis platform [31]. Default settings were used for quality control, de-replication, OTU clustering, and classification at a 97% sequence identity level. Taxonomic assignments were based on the SILVA database release 123 (24 July 2014).

Contigs were generated for dominant amplicons to allow for phylogenetic reconstruction and T-RF prediction as previously described [27,28]. Briefly, contigs were assembled from matching forward and reverse reads with the SEQMAN II software (DNAStar) using assembly thresholds of at least 98% sequence similarity over a 50-bp match window. Contigs without at least one forward and one reverse read were not considered for further analysis. All pyrosequencing reads generated in this study have been deposited in NCBI’s Sequence Read Archive and are accessible as SRA study SRP004457. Selected assembled amplicon contigs of dominating populations have been deposited with GenBank under the accession numbers HQ596373 to HQ596401.

### 2.3. Data Handling and Statistics

Shannon diversity (*H*′) and functional organization (*Fo*) of communities were inferred for OTUs obtained from rarefied sequencing libraries (*n* = 1200 processed reads for each). *Fo* is an evenness indicator based on Pareto-Lorenz curves [32]. As in *H′*, community richness and relative abundances of individual taxa are incorporated. However, rare taxa are less important, as only the cumulative relative abundance of the top 20% of the most abundant OTUs is considered. *Fo* would be 0.2 at perfect evenness; the higher the *Fo*, the more organized, less diverse, and less even the respective community.

Canonical correspondence analysis (CCA) was used to examine the assembly of taxonomic groups along gradients of environmental variables [33] measured across the plume. CCA was performed in R (version 3.3.2) using the vegan package [34]. The significance of CCA axes and explanatory variables was assessed by permutation tests with 1000 permutations using the ‘anova.cca’ function. Variables were selected by backward selection starting with a full model containing toluene concentration, sulfide concentration, depth below surface, *bssA* gene copy numbers, and groundwater level. Only significant variables (*p* < 0.05) were retained in the final model. Principal component analysis (PCA) was done using the ‘prcomp’ function implemented in R to analyze changes in the relative abundances of taxonomic groups over the years for individual plume zones (upper and lower plume fringe, deeper zone).

## 3. Results

### 3.1. Hydraulic Dynamics and Plume Fluctuation

Considerable dynamics of the groundwater table (GWT) were observed at the site during four years of sampling (Figure 1). A drop of the GWT of almost 40 cm between July 2006 and January 2007 was followed by a constant rise of ~60 cm until March 2009. After this, an equivalent but more rapid decline of the water table was observed until winter 2009.

Between the samplings in February 2006 and February 2007, the drop of the water table was reflected in a downshift of the toluene plume core from ~6.6 to 6.8 m depth, while maximal concentrations remained at ~40 mg·L^−1^ (Figure 2a). The reactive lower fringe of the plume remained localized at ~7.1 m depth (similar to the lower fringe reported by Winderl et al. [12] for 2005) and reactive, as suggested by an increase of δ^13^C ratios of toluene from approximately −24‰ to −21‰ in this zone (Figure 2b). Maximal sulfide concentrations at the lower fringe, however, decreased from ~10 to ~2 mg·L^−1^ over the same time (Figure 2c). In September 2008, after the rise of the GWT, a corresponding upshift of the plume core was paralleled by a doubling of maximum toluene concentrations to ~100 mg·L^−1^, now at ~6.5 m depth. A peak of sulfide concentrations of ~10 mg·L^−1^, now at 6.9 m depth, indicated a stimulation and upshift of sulfidogenesis. However, a loss of notable δ^13^C fractionation was observed for toluene at the plume fringes in 2008 (Figure 2b). Instead, overall δ^13^C ratios of toluene appeared to level out at slightly increased ratios (−23‰ to −22‰) over the plume. In June 2009, maximal toluene concentrations in the plume core dropped markedly, to ~18 mg·L^−1^ now at 6.6 m depth (Figure 2a). Maximal sulfide concentrations of ~6 to ~7 mg·L^−1^ were observed at both the upper (6.4 m depth) and lower (6.9 m) plume fringes, connected to a partial re-establishment of toluene δ^13^C isotope fractionation, however, mostly at the upper plume fringe (−20.5‰ at 6.4 m depth).

The horizontal extent and localization of the Flingern BTEX plume, as monitored by the site owner, remained stable over the whole period (Appendix A). The HR-MLW was always centrally placed in the transect of the plume over the years, and pronounced lateral fluctuations of the plume or groundwater flow were not observed. At the same time, the concentration of other, less abundant contaminants also detectable at the site (e.g., benzene, naphthalene) remained constant or even increased in concentration in the plume core between 2008 and 2009 (Appendix A), thus not reflecting the marked drop in toluene concentrations over the same time.

### 3.2. Dynamics of Plume Microbiota

Sediment bacterial 16S rRNA gene counts (maximally ~2 × 10^8^ in the plume core in 2005 and 2006) dropped by an order of magnitude throughout the plume core in 2008, but were still on a level comparable to the qPCR results of 2005 (~2 × 10^7^) at the lower fringe (Figure 3a). In 2009, the depth profile of bacterial 16S rRNA gene counts was again comparable to that before 2008. Anaerobic toluene degraders were quantified via a qPCR assay established for the Desulfobulbaceae-related benzylsuccinate synthase α-subunit (*bssA*) genes of previously dominating degraders at the site [12,27]. Compared to the maximum catabolic gene counts of >10^7^ per gram of sediment observed for the lower plume fringe in 2005 [12] and also in 2006 (Figure 3b), a decrease of this gene pool by over an order of magnitude was observed in 2008. Still, the ratio of Desulfobulbaceae-related *bssA* genes to total bacterial 16S rRNA gene counts continued to be highest at the lower plume fringe (Figure 3c). However, gene enrichment was clearly higher (~0.5 to 0.8) before than after (~0.1 to 0.2) the hydraulic dynamics between 2007 and 2008.

High-throughput sequencing of 16S rRNA gene amplicons of sedimentary bacteria was performed for more detailed structural insights into the Flingern plume microbiota. For the present study, amplicon libraries from five depths over the plume transect taken in 2006, 2008, and 2009 were sequenced. The libraries contained an average of ~2700 reads per sample at a threshold of >250 bp quality-trimmed read length.

Bacteria at the upper plume fringe were dominated by members of the Alpha-, Beta-, and Gammaproteobacteria, as well as by Acidobacteria (Figure 4). Between the three sampling time points, abundance shifts were mostly observed for reads related to *Thiobacillus* spp., which was abundant (up to ~10% of total reads) in 2006 and 2009, but almost not detectable after the rise of the GWT in 2008. In contrast, diverse unclassified Gammaprotobacteria appeared especially abundant in 2008 (~33%). An increase of sequences within the Acidobacteria (~20%), many related to *Geothrix* spp. (~7%), was observed at the upper fringe in 2009.

Bacteria at the lower plume fringe were always dominated by reads within the Deltaproteobacteria, Betaproteobacteria, and Clostridia (Figure 4). Within the Deltaproteobacteria, sequencing reads related to *Geobacter* spp. were stable in abundance (~18–20%) over the years, but almost absent in other depths. In contrast, a high abundance of reads within the Desulfobulbaceae (~30% and 42% in 2006 and 2008, respectively) decreased to only ~9% in 2009. Vice versa, reads related to *Desulfosporosinus* spp. within the Peptococcaceae (Clostridia) constantly increased in abundance, from only ~8% in 2006 to ~26% in 2009. The diversity of rarefied amplicon pools was always lowest at the lower plume fringe (Appendix A), consistent with a peaking ‘functional organization’ (*Fo*) of these communities (Appendix A).

Finally, amplicon pools from the deeper zone revealed distinct, but much more stable communities over the years. They were also dominated by Deltaproteobacteria and Clostridia, but also by Chloroflexi (~11%). The Deltaproteobacteria from the deeper zone, in contrast to those recovered at the lower plume fringe, were mostly affiliated to the Desulfobacteraceae (up to 23%), as well as to other unclassified Deltaproteobacteria. High ratios (up to 38%) of bacterial reads unclassified at the phylum level were also found at these depths.

We have previously demonstrated a high reproducibility and semi-quantitative accuracy of relative OTU abundances across biologically replicated sequencing libraries for a series of triplicate DNA extracts from sediments taken from the plume core (~6.8 m depth) in the same sampling campaign [28]. Still, as the community shifts observed for the different time points and depths in the present study resulted from non-replicated sequencing libraries, we aimed to confirm the most important population dynamics via biologically replicated T-RFLP fingerprinting. For this, amplicon contigs of dominating degrader populations at the lower plume fringe were assembled and corresponding T-RFs were predicted (Appendix A). The results were consistent with previous T-RF assignments for populations at the site [12,13,27,28,36]. As expected, degrader populations within the Desulfobulbaceae, the Geobacteraceae, and the Peptococcaceae were affiliated to the 159, the 129, as well as the 138 and 228 bp T-RFs, respectively (Appendix A). While the abundance profile of the 129 bp fragment (*Geobacter* spp.) was rather stable over time, maximal abundance of the 159 bp T-RF (Desulfobulbaceae) was consistently reduced at the lower plume fringe in June 2009 (from ~45% to ~15% abundance at ~7.1 m depth). In contrast, the signal of the combined 138 and 228 bp T-RFs (*Desulfosporosinus* spp.) more than doubled (from ~10% to ~30% at 7.1 m depth) in the same year (Appendix A).

### 3.3. Multivariate Statistics of Degrader Dynamics

Multivariate statistics substantiated the marked distinctions between sequencing OTUs detected in samples from the different depth zones respectively, as well as their association to signifying environmental variables (Figure 5a). At the upper fringe, especially the abundance of Alphaproteobacteria, *Pseudomonas*, and *Thiobacillus* spp. correlated with high toluene concentrations and shallow depth. In contrast, *Desulfosporosinus* and *Sedimentibacter* spp. within the Clostridia, as well as Desulfobulbaceae and *Geobacter* spp., showed a strong association with the lower fringe samples, intermediate toluene concentrations, and high *bssA* copy numbers. *Desulfobacterium* spp. and other Deltaproteobacteria, in addition to diverse Clostridia, were resolved as characteristic for the deeper zone below the plume, showing strong negative correlations with toluene while positive correlations with depth.

Although the CCA clearly supported distinct niches for OTUs across the plume, the analysis was not sufficient to resolve dynamics in abundances of single taxonomic groups within different zones over the years. Therefore, we analyzed changes in community composition that occurred over time by PCA for each individual plume zone. For the upper fringe (Figure 5b), dynamics in community composition appeared most strongly driven by abundance changes of Gammaproteobacteria other than *Pseudomonas* spp. in 2008 (contributing 69% to PC1), and Acidobacteria (contributing 60% to PC2), which strongly increased in 2009. Community dynamics at the lower fringe (Figure 5c) were mainly associated to changes in abundance of the Desulfobulbaceae (contributing 73% to PC1) and *Desulfosporosinus* spp. (contributing 14% and 21% to PC1 and PC2, respectively). A sharp negative correlation of Desulfobulbaceae was found to PC1 and especially the 2009 sample. In contrast, *Desulfosporosinus* spp. exhibited an opposed pattern and a strong negative correlation with the Desulfobulbaceae. In addition, the Comamonadaceae and diverse Betaproteobacteria (contributing 31% and 32% to PC2, respectively) had positive correlations to the 2006 sample. Community dynamics in the deeper zone (Figure 5d) appeared mainly driven by a decrease of diverse Clostridia (contributing 23% to PC1) especially between 2006 and 2008, and a concomitant increase of diverse Deltaproteobacteria (contributing 59% to PC1). However, in contrast to the plume fringes, species abundances only changed marginally after 2008: the sample points for 2008 and 2009 were mainly separated by PC2, which only accounted for a small fraction (12.6%) of the explained variance.

## 4. Discussion

### 4.1. Hydraulic Dynamics and Biodegradation

This field study demonstrates that hydraulic dynamics of a contaminant plume can be paralleled by dynamics of biogeochemical processes, pollutant degradation, and plume microbiota in situ. It should be cautiously discussed, however, whether the observed biogeochemical dynamics of the plume were essentially linked to the fluctuations of the water table, or possibly also to other driving mechanisms. A number of previous studies have demonstrated that feedbacks between recharge processes, seasonal water table fluctuations, and biodegradation processes actually exist in porous media [17,18,19,37]. However, it is also well established that seasonal variability of groundwater flow direction can profoundly influence the measurement of contaminants and redox species at defined sampling locations [38,39]. In our case, long-term monitoring data of groundwater flow by the site owner as well as the continued centric localization of the HR-MLW within the lateral plume transect (Appendix A) suggest that variable groundwater flow directions were not a major contributor to biogeochemical variability of vertical plume transects observed over this study.

Thus, an impact of vertical hydraulic fluctuations (Figure 1) on the plume system seemed at least probable. However, even without assuming direct feedbacks on the microbes, water table fluctuations can be expected to impact contaminant loads and redox compartments in plumes [39]. Complex and spatially variable source zones of hydrocarbons exist in the subsurface of many contaminated sites, including in Flingern. Thus, a rise of the groundwater table could have increased the contribution of a secondary hydrocarbon source zone to the plume, which could have caused the increase in maximal toluene concentrations in 2008. The observed leveling-out of toluene δ^13^C isotope ratios at increased ratios in the same year (Figure 2b) was also consistent with the possible intrusion of a secondary contaminant source [40]. Nevertheless, the loss of toluene stable isotope fractionation at the plume fringes in 2008 remains a strong indicator for decreased biodegradation activities [41]. The stimulated sulfidogenesis observed in the same year could either have been connected to oxidation processes other than that of toluene, or to the establishment of a secondary toluene degrader population with a less pronounced isotope fractionation pattern. Consistently, carbon and hydrogen stable isotope studies have reported distinct enrichment factors for a number of anaerobic degraders of BTEX hydrocarbons within the Deltaproteobacteria, Betaproteobacteria, and Clostridia [42,43]. In summary, our biogeochemical data suggests that the pronounced dynamics of contaminant distribution and biodegradation processes observed at the site were, at least in part, attributed to hydraulic dynamics during our study.

### 4.2. Dynamics of Plume Microbiota

At the Flingern site, the importance of the lower plume fringe for anaerobic toluene degradation and a corresponding enrichment of degrader populations within the Desulfobulbaceae have been previously demonstrated [12,13]. Thus, it was not surprising that also in the present study, the lower fringe always carried the highest absolute and relative abundance of *bssA* genes (Figure 2), as well as the lowest diversity and highest functional organization of degrader communities (Appendix A). This was in line with the concept of a high specialization of the respective microbial community [44]. DNA-stable isotope probing (SIP) experiments with lower fringe sediments taken in 2008 have previously proven the affiliation of the quantified *bssA* genes to the detected Desulfobulbaceae, and verified an important role of this population in toluene degradation under sulfate reduction in situ [27]. Prior to the plume dynamics after 2006, these Desulfobulbaceae continued to dominate the degrader community at the lower fringe. Closely related Desulfobulbaceae have also been identified as sulfidogenic toluene degraders in other contaminated terrestrial subsurface systems [45,46,47] and seem of considerable environmental importance [11]. The most closely related available isolate, *Desulfoprunum benzoelyticum*, albeit not yet tested for aromatics, has been shown to degrade benzoate [48], a central intermediate of anaerobic aromatics degradation.

After the dynamics of the plume, *Desulfosporosinus* spp. populations increased in abundance and appeared to replace the previously dominating Desulfobulbaceae at the lower plume fringe. This abundance switching was not only observed in sequencing libraries, but consistently found in biologically replicated T-RFLP fingerprinting (Appendix A) and supported also by multivariate statistics (Figure 5). This demonstrates the utility of combined sequencing and fingerprinting datasets in environmental studies [49]. Members of the genus *Desulfosporosinus* are also known as sulfate-reducing toluene degraders in terrestrial systems [46,50,51], and have also been found at the Flingern site [27]. At the same time, other members of the lower fringe community remained relatively stable in abundance over time (e.g., the Geobacteraceae), suggesting that community shifts occurred specifically for sulfate-reducing toluene degraders.

The establishment of *Desulfosporosinus* spp. degraders was not apparent on a catabolic gene level, as *bssA* ratios remained low in 2009 (Figure 3c). However, the employed qPCR assay specifically targets the Desulfobulbaceae ‘F1-cluster’ *bssA* genes originally dominating at the site [12,35]. While more advanced primer sets capable of targeting clostridial *bssA* genes are now available [52], the detection of this gene lineage was not conducted in the present study.

It must be cautiously discussed whether the hydraulic and biogeochemical dynamics of the plume could indeed have been connected to the observed switching in abundant degrader populations at the lower fringe. Specific ecological features of the Gram-positives, such as spore formation, could have been involved in an improved resistance to and recovery of these populations after hydraulic forcing. Repeated stress and reactivation cycles are amongst the prime mechanisms proposed to maintain microbial diversity and functional redundancy in natural habitats [53]. However, as our observations come from a complex field setting, a direct coupling between hydraulic and microbial habitat dynamics cannot be unambiguously proven. This clearly calls for a more controlled experimental verification under laboratory conditions. Nevertheless, hydraulically induced plume shifts could represent an under-regarded deterministic factor controlling aquifer microbial community assembly and biogeochemical activities in contaminated systems. Similar feedback mechanisms have also been revealed for hydraulically driven solute and nutrient fluxes at a surface water–groundwater interface [54,55]. In unconfined porous aquifers, water table fluctuations are directly connected to dynamic recharge regimes. Since the frequency and intensity of precipitation extremes is predicted to increase in the near future [56], a better understanding of the consequences for aquifer biogeochemistry and groundwater quality may be urgently needed.

### 4.3. Conclusions

In summary, our results provide evidence that complex couplings exist between hydraulic forcing, degrader community dynamics, and contaminant degradation in situ. Functional redundancy amongst anaerobic hydrocarbon degrader populations might have been relevant in sustaining biodegradation after habitat dynamics. Hence, contaminant plumes should be more comprehensively investigated as a habitat of dynamic microbial communities, both in controlled laboratory systems, but also in the field. This is relevant for established monitoring and also management strategies for contaminated sites, where spatially explicit and fine-scale plume transects are still rarely considered [11,57]. We suggest that focusing not only on degrader abundance, but also on degrader diversity and dynamics provides an improved ecological understanding of contaminated groundwater systems, where feedbacks between abiotic and biotic ecosystem components are largely under-regarded.

## Figures and Tables

**Figure 1 microorganisms-07-00046-f001:**
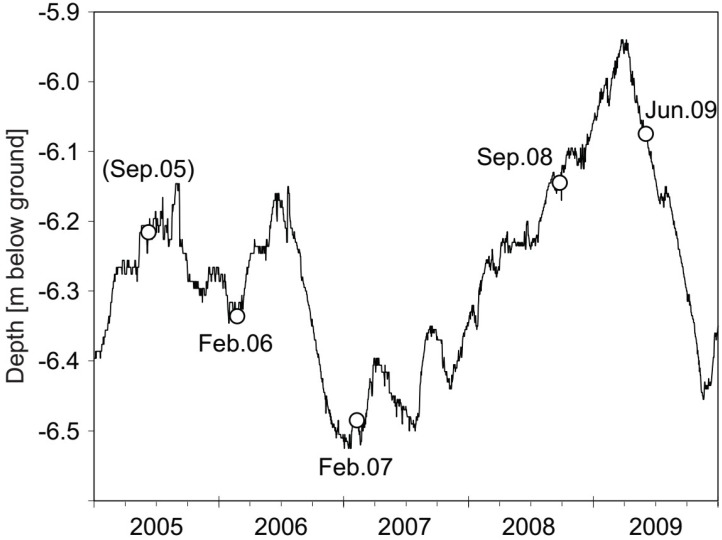
Fluctuation of the groundwater table over the period of investigation. Measurements were performed using water table logging in the conventional monitoring wells 19179 and 19223 (Appendix A) adjacent to the high-resolution multi-level well (HR-MLW) used for high-resolution water and sediment sampling. Circles indicate the four time points of sediment and groundwater sampling, and one time point of groundwater sampling only (February 2007). Field data from September 2005 have been previously reported [12].

**Figure 2 microorganisms-07-00046-f002:**
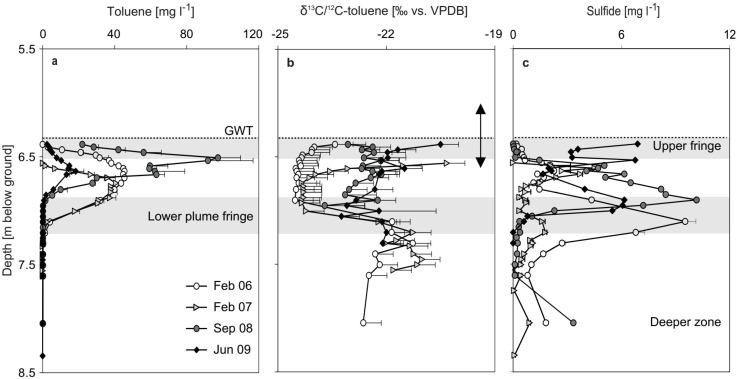
Depth profiles of toluene concentration, biodegradation, and sulfide in the Flingern aquifer over four successive sampling time points. (**a**) Toluene concentrations plus standard deviation (positive only); (**b**) toluene stable isotope ratios plus standard deviation (positive only); (**c**) sulfide concentrations plus standard deviation (positive only) from duplicate groundwater samples per time point and depth. The extent of groundwater table (GWT) fluctuations and shifts of the plume boundaries during the study period are indicated as a vertical arrow (at GWT). The localization of the upper and lower plume fringes is highlighted in grey, but averaged over the entire study period.

**Figure 3 microorganisms-07-00046-f003:**
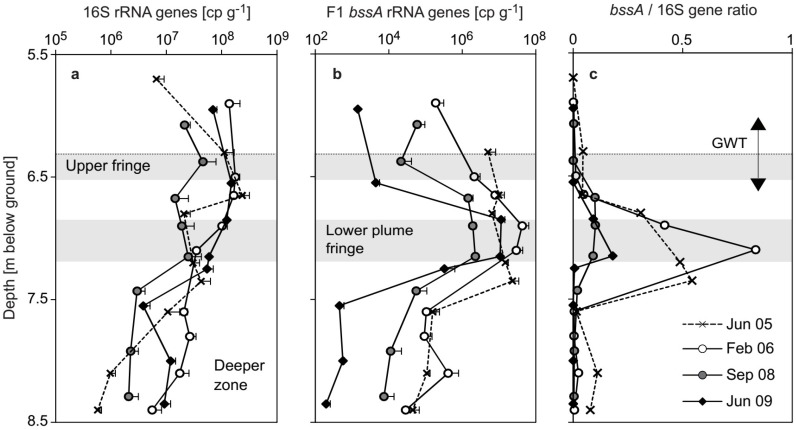
Depth profiles of sedimentary bacterial rRNA genes and anaerobic toluene degradation genes at the Flingern aquifer during the study. (**a**) Total bacterial 16S rRNA genes and (**b**), ‘F1-cluster’ *bssA* genes [12,35] as quantified via qPCR of triplicate sediment DNA extracts. Shown are average gene counts g^−1^ sediment plus standard error (positive only). (**c**) Relative *bssA* gene abundance within total gene counts. Data from June 2005 were already reported previously [12]. Further details are as in Figure 2.

**Figure 4 microorganisms-07-00046-f004:**
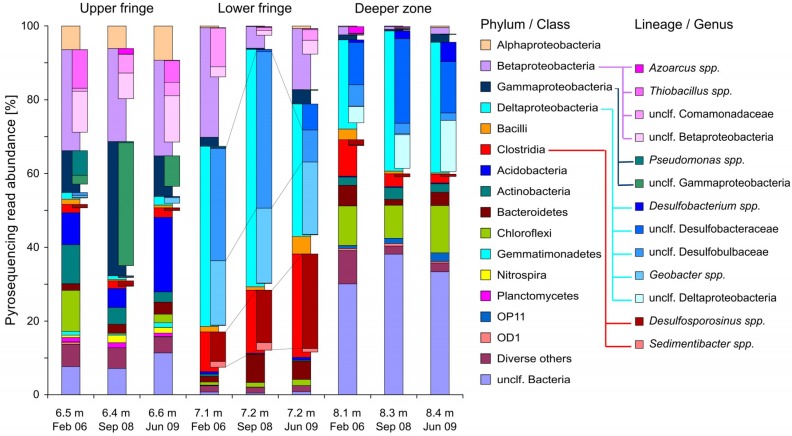
Temporal bacterial community dynamics in compartments of the Flingern hydrocarbon plume as revealed by amplicon pyrosequencing of 16S rRNA gene fragments. Total communities are resolved to phylum/class-level, while selected dominating and dynamic lineages are highlighted at the genus-level.

**Figure 5 microorganisms-07-00046-f005:**
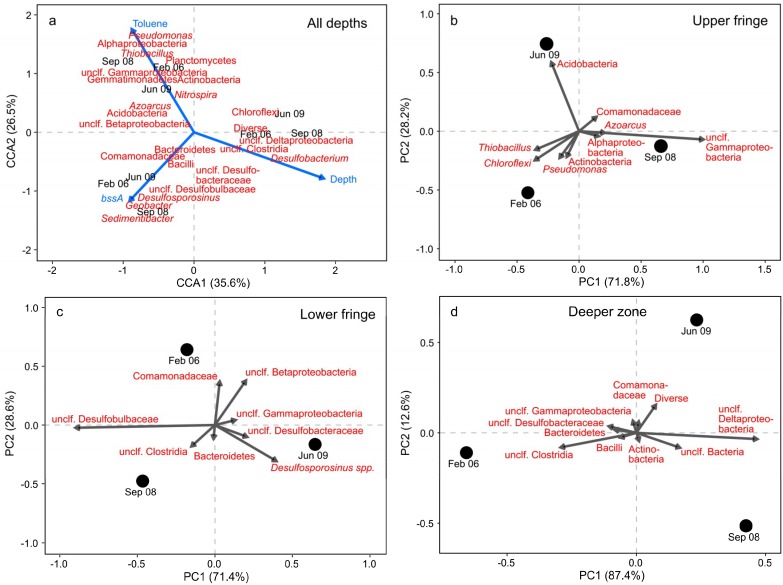
(**a**) Canonical correspondence analysis (CCA) triplot of overall samples (black), sequencing OTUs (red), and explanatory variables (blue) analyzed in this study. The proportion of inertia explained by the constrained model was 66.1% of which 62.1% was retained in the first two axes. Significance of CCA axes and explanatory variables (1000 permutations): CCA1 (*p* = 0.013); CCA2 (*p* =0.035); toluene concentration (*p* = 0.009), depth below surface (*p* = 0.008), *bssA* gene copy numbers (*p* = 0.025). (**b**–**d**) PCA biplots showing changes of relative OTU abundances for individual plume zones over the years. All OTUs resolved in sequencing were included in statistical analysis, but for clarity of display, only vectors of OTUs with a combined loading of >1% on the first two principle components are shown.

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
