# Peer review of "Dynamics of Hydrology and Anaerobic Hydrocarbon Degrader Communities in A Tar-Oil Contaminated Aquifer"

_microorganisms, 2019, doi:10.3390/microorganisms7020046_

Round 1
Reviewer 1 Report
The paper presents interesting multi-annual research. It is written in comprehensible and correct language.
However, there are some inaccuracies related to the NGS sequencing methodology. The authors confuse the sequencing based on 16S rRNA amplicons with transcriptomics. It cannot be said that transcriptomics or transcriptomes have been used, while the method is based on DNA isolation, DNA amplification and DNA sequencing. However, I believe that the error occurred at the stage of describing the sequence when uploading to publicly available databases and does not affect the results obtained, but it does mislead the reader and should be corrected.
It would be more convenient to read the paper if the methodology described more extensively in this paper, instead of referring readers to their own previous papers (high level of self-citation). I understand that such a procedure makes the work seem concise, more straightforward and avoids duplication of the same text, but it loses its reading value.
The following remarks do not impair the positive value of the manuscript:
Section: 2.2 Molecular analyses
ln 106 - 127:
- Please describe more precisely the DNA extraction method.
- How many subsamples were taken
- Why qPCR was performed only on DNA material, not the mRNA (it could better reflect if chosen genes were actively involved in toluene degradation)
- the methodology for the sequencing and bioinformatics analysis is given as an external source. Please provide at least the name and sequences of the primers used. It is very uncomfortable to read the manuscript, in which the authors often refer to other/previous papers.
- which version of mothur was used
- why 97% similarity cutoff was chosen, while the current standard is at least 99% . Using 97% some different species will be undistinguishable. Previously, 97% were used because the computation limits, which now is not an issue.
- primer sequences were removed from reads?
- there are some inconsistencies: sequencing was performed on the 16S rRNA amplicon library, but the description on the GEO NCBI database says that the library source = transcriptomic and the description says “small rRNA were amplified from total nucleic acids extracted from environmental samples using specific PCR primers”. I assume that you have isolated total DNA (not RNA) and amplify DNA fragments (not RNA, again) of 16S rRNA gene region. So there is no transcription since you isolate DNA, amplify DNA and sequence DNA.
Otherwise you should isolate total RNA, perform reverse transcription (RNA -> cDNA) and then you can say about “transcriptiomics”.
- again, looking at SRA NCBI portal (e.g. your SRX031255 submission), there is a mistake (same as above). You didn’t performed transcriptomics.
- Greengenes database is outdated and no longer recommended.
Section: 3.2
Fig.4 – It would be more readable if the results were presented in separate charts. Separate presentation of genera would simplify the comparison. It would be also interesting to present the total number of identified genera.
Author Response
REV1
The paper presents interesting multi-annual research. It is written in comprehensible and correct language.
However, there are some inaccuracies related to the NGS sequencing methodology. The authors confuse the sequencing based on 16S rRNA amplicons with transcriptomics. It cannot be said that transcriptomics or transcriptomes have been used, while the method is based on DNA isolation,
DNA amplification and DNA sequencing.
– The entries in GEO were edited to better reflect that DNA and not RNA was sequenced, an update to the sample analysis procedure was posted, as well as updates to the fields “Library strategy”, “Library source” and “Library selection” (now updated as “other”).
– The SRA entries were edited to reflect that the sequences are not from “transcriptomics” and the fields “Strategy” were updated as “genomic”. We hope that these changes are satisfactory to the reviewer.
However, I believe that the error occurred at the stage of describing the sequence when uploading to publicly available databases and does not affect the results obtained, but it does mislead the reader and should be corrected.
It would be more convenient to read the paper if the methodology described more extensively in this paper, instead of referring readers to their own previous papers (high level of self-citation). I understand that such a procedure makes the work seem concise, more straightforward and avoids duplication of the same text, but it loses its reading value.
The following remarks do not impair the positive value of the manuscript:
Section: 2.2 Molecular analyses
ln 106 - 127:
- Please describe more precisely the DNA extraction method.
- How many subsamples were taken
- Why qPCR was performed only on DNA material, not the mRNA (it could better reflect if chosen genes were actively involved in toluene degradation)
- the methodology for the sequencing and bioinformatics analysis is given as an external source. Please provide at least the name and sequences of the primers used. It is very uncomfortable to read the manuscript, in which the authors often refer to other/previous papers.
- which version of mothur was used
- why 97% similarity cutoff was chosen, while the current standard is at least 99% . Using 97% some different species will be undistinguishable. Previously, 97% were used because the computation limits, which now is not an issue.
In fact, we differentiate and interpret mainly at the family level, and only touch on the genus level for a few selected populations. Finally, minor differences in ecological interpretation are usually reported when comparing 97 and 99% cutoffs, apart from increased abundance of singletons (e.g., Poretsky et al., PLoS One. 2014; 9(4): e93827). Therefore, we hope that these arguments can convince the reviewer of the merits of using a 97% cutoff in our present study.
- primer sequences were removed from reads?
- there are some inconsistencies: sequencing was performed on the 16S rRNA amplicon library, but the description on the GEO NCBI database says that the library source = transcriptomic and the description says “small rRNA were amplified from total nucleic acids extracted from environmental sample susing specific PCR primers”. I assume that you have isolated total DNA (not RNA) and amplify DNA fragments (not RNA, again) of 16S rRNA gene region. So there is no transcription since you isolate DNA, amplify DNA and sequence DNA.
Otherwise you should isolate total RNA, perform reverse transcription (RNA -> cDNA) and then you can say about “transcriptiomics”.
- again, looking at SRA NCBI portal (e.g. your SRX031255 submission), there is a mistake (same as above). You didn’t performed transcriptomics.
- Greengenes database is outdated and no longer recommended.
Section: 3.2
Fig.4 – It would be more readable if the results were presented in separate charts. Separate presentation of genera would simplify the comparison. It would be also interesting to present the total number of identified genera.
Reviewer 2 Report
This is an interesting study where the authors investigated the effects of hydraulic dynamics on anaerobic degrader communities in an oil-contaminated aquifer. The manuscript is well written and was easy to follow. My major concern is the way how the impacts of hydraulic dynamics on microbial communities were analyzed and discussed. I did not get a complete sense of the effects of hydraulic variations on the degrader community; the results rather indicate more of temporal variations within a fixed plume boundary. The hypothesis needs to be specific with more context on functional redundancy in a dynamic hydraulic environment (see my specific comment below). Supplementary figures were cited in several places, which may distract readers, and some of these can be included in the main text. I recommend changing the focus of this study more towards the temporal dynamics (inter-annual variations) in plume biogeochemistry rather than directly relating this to hydraulic dynamics unless more experimental evidence are added.
Introduction
L56: Elaborate more on the ‘reactive plume-fringe’ concept;
L68: cite the previously characterized study
L68-69: Be more specific about what specific attributes of the degrader communities are expected to change and why. Also, the idea of ‘functional redundancy’ and its relationship with hydraulic dynamics are not introduced.
L71-72: Move to the conclusions
Materials and methods
Add more details on sampling design such as number of replicates, distance between the sampling points, and abiotic environment of the plumes.
L92-93: Is different sampling time a concern for detecting the localized centric zones of plumes? How did other groundwater recharge phenomena affect this?
L104: Briefly describe how these were done.
Results
May be add a section to characterize the plume boundaries (upper, lower, and deeper) and nature of contaminations.
Fig. 2 and 3: Did the plume boundaries (upper and lower ends) not change in different years? These figures only show temporal variation in BTEX and microbiome data.
L170: Rephrase
L257-263: Can different zones (upper, lower, and deeper) be indicated in this graph?
L273-274: Why then PCA was not used for the pooled ‘All depths’ graph?
Discussion
L302-305: This is not clear from Fig. S1. The 2009 figure shows spatial variability in BTEX concentrations. The other two figures (2006 and 2007) just show contour lines. Also, what is the experimental evidence that the variability in spatial distribution of plumes does not impact the vertical biogeochemistry; specially when this was not something measured in other sampling locations!
L331: Is ‘prior to the plume dynamics’ indicating the sampling done before the fluctuating hydraulic phenomena?
L360-363: Major limitation of the study! The idea of ‘plume shift’ is introduced here but all the figures show a fixed plume boundary!
Author Response
REV2
This is an interesting study where the authors investigated the effects of hydraulic dynamics on anaerobic degrader communities in an oil-contaminated aquifer. The manuscript is well written and was easy to follow. My major concern is the way how the impacts of hydraulic dynamics on microbial communities were analyzed and discussed. I did not get a complete sense of the effects of hydraulic variations on the degrader community; the results rather indicate more of temporal variations within a fixed plume boundary.
The hypothesis needs to be specific with more context on functional redundancy in a dynamic hydraulic environment (see my specific comment below). Supplementary figures were cited in several places, which may distract readers, and some of these can be included in the main text. I recommend changing the focus of this study more towards the temporal dynamics (inter-annual variations) in plume biogeochemistry rather than directly relating this to hydraulic dynamics unless more experimental evidence are added.
Although we strongly oppose the reviewer’s perception of a static plume boundary, which was clearly not the case over the years, the reviewer is fully correct that a final mechanistic conclusion is not possible without further direct experimental evidence. However, such evidence could only be provided via a controlled experimental setting, and not from field monitoring where a fully reliable control is essentially not possible. As also this was already explained in the original manuscript (L405, L419), we have now hope the reviewer can reconsider our very cautious lines of argumentation, and can accept that we prefer to not rewrite the manuscript with a focus on biogeochemical dynamics alone.
Introduction
L56: Elaborate more on the ‘reactive plume-fringe’ concept;
L68: cite the previously characterized study
L68-69: Be more specific about what specific attributes of the degrader communities are expected to change and why. Also, the idea of ‘functional redundancy’ and its relationship with hydraulic dynamics are not introduced.
L71-72: Move to the conclusions
Materials and methods
Add more details on sampling design such as number of replicates, distance between the sampling points, and abiotic environment of the plumes.
L92-93: Is different sampling time a concern for detecting the localized centric zones of plumes? How did other groundwater recharge phenomena affect this?
L104: Briefly describe how these were done.
Results
May be add a section to characterize the plume boundaries (upper, lower, and deeper) and nature of contaminations.
Fig. 2 and 3: Did the plume boundaries (upper and lower ends) not change in different years? These figures only show temporal variation in BTEX and microbiome data.
We can only speculate that the reviewer has come to this impression by the fact that the shaded area of the lower plume fringe is averaged over the years in Figs. 2 and 3. We see no other way to more optimally present the comprehensive data in figures summarizing the data over the years.
However, we would like to stress that the legend of the figures clearly explains that “the extent of the fluctuations … are indicated (vertical arrow at GWT)” and that “The zones of the upper and lower plume fringes are averaged over the entire study period…”. We hope that the reviewer can accept this line of argumentation and presentation. We have now added further detail to the figure legends to better resolve these issues.
L170: Rephrase
L257-263: Can different zones (upper, lower, and deeper) be indicated in this graph?
L273-274: Why then PCA was not used for the pooled ‘All depths’ graph?
Discussion
L302-305: This is not clear from Fig. S1. The 2009 figure shows spatial variability in BTEX concentrations. The other two figures (2006 and 2007) just show contour lines. Also, what is the experimental evidence that the variability in spatial distribution of plumes does not impact the vertical biogeochemistry; especially when this was not something measured in other sampling locations!
Clearly, there was variation in BTEX concentrations over the years, and we never claimed that these were not. The point we would like to make here is simple: The HR-MLW was always centrally placed within the plume! We believe this is sufficiently documented by this supplementary figure.
To the second aspect, we are fully aware that we have no experimental evidence. This could only be provided via a laboratory microcosm experiment, which would clearly be beyond the scope of the present manuscript. We feel that we adequately address this limitation and inspiration for future work in the discussion (L405, L419).
L331: Is ‘prior to the plume dynamics’ indicating the sampling done before the fluctuating hydraulic phenomena?
L360-363: Major limitation of the study! The idea of ‘plume shift’ is introduced here but all the figures show a fixed plume boundary!